# Genetic variants associated with sepsis

**Milo Engoren**[1]*, **Elizabeth S. Jewell**[1], **Nicholas Douville**[1], **Stephanie Moser**[1], **Michael D. Maile**[1], **Melissa E. Bauer**[1,2]

**1** Department of Anesthesiology, University of Michigan, Ann Arbor, MI, United States of America,
**2** Department of Anesthesiology, Duke University, Durham, NC, United States of America

* engorenm@med.umich.edu

## Abstract

### Background

The variable presentations and different phenotypes of sepsis suggest that risk of sepsis comes from many genes each having a small effect. The cumulative effect can be used to create individual risk profile. The purpose of this study was to create a polygenic risk score and determine the genetic variants associated with sepsis.

### Methods

We sequenced ~14 million single nucleotide polymorphisms with a minimac imputation quality $R2 > 0.3$ and minor allele frequency $> 10^{-6}$ in patients with Sepsis-2 or Sepsis-3. Genome-wide association was performed using Firth bias-corrected logistic regression. Semi-parsimonious logistic regression was used to create polygenic risk scores and reduced regression to determine the genetic variants independently associated with sepsis.

### Findings

2261 patients had sepsis and 13,068 control patients did not. The polygenic risk scores had good discrimination: c-statistic = $0.752 \pm 0.005$ for Sepsis-2 and $0.752 \pm 0.007$ for Sepsis-3. We found 772 genetic variants associated with Sepsis-2 and 442 with Sepsis-3, $p < 0.01$. After multivariate adjustment, 100 variants on 85 genes were associated with Sepsis-2 and 69 variants in 54 genes with Sepsis-3. Twenty-five variants were present in both the Sepsis-2 and Sepsis-3 groups out of 32 genes that were present in both groups. The other 7 genes had different variants present. Most variants had small effect sizes.

### Conclusions

Sepsis-2 and Sepsis-3 have both separate and shared genetic variants. Most genetic variants have small effects sizes, but cumulatively, the polygenic risk scores have good discrimination.

**Data Availability Statement:** Data are available from the Michigan Genomics Initiative https://precisionhealth.umich.edu/our-research/michigangenomics/#request.

**Funding:** The authors received no specific funding for this work.

**Competing interests:** Nicholas Douville was supported by a Foundation for Anesthesia Education and Research (FAER) Mentored Research Training Grant Michael Maile was supported by the American Diabetes Association. Milo Engoren has received consulting fee from Aerogen and Masimo. The other authors have nothing to declare. This does not alter our adherence to PLOS ONE policies on sharing data and materials.

**Abbreviations:** GWAS, Gene wide association studies; MGI, Michigan Genomics Initiative; SIRS, Systemic Inflammatory Response Syndrome; SNP, single nucleotide polymorphisms; SOFA, Sequential Organ Failure Assessment.

## Background

Studies show that sepsis is a leading cause of mortality, the incidence is increasing, and survivors frequently have long-term physical, psychological, and cognitive impairments [1–4]. Sepsis is a syndrome, previously described by the presence of markers of systemic inflammation–abnormal vital signs and leukocyte counts—associated with suspected infection. Recently, the definition has changed to one that relies on organ system dysfunction [5–8]. Sepsis is treated with antibiotics, intravenous fluids, vasopressors and supportive care. Attempts to decrease mortality in sepsis via drugs that target specific enzymes, mediators, or proteins associated with sepsis have invariably failed, likely because sepsis has multiple pathways involving many enzymes, mediators, and proteins coded by a plethora of genes, making sepsis a polygenic disease, like hypertension [9, 10].

Sepsis and the immune response to infection have a genetic component–about 10% of human genes codes for immune mediators [11]–and adult adoptees have nearly a five-fold higher risk of infection-related mortality if a biologic parent died of infection [12]. This is a higher heritability than cardiovascular disease or cancer [12]. Studies have implicated many genes across a large spectrum of immune and coagulation proteins including interleukins, receptors, and fibrinogen [13]. However, many of these studies were limited by relatively small patient populations, which may limit the reproducibility of the findings. More recently, genome-wide association studies (GWAS) have been performed on patients with pneumonia or with severe sepsis [14, 15]. These studies found a few associated genes, in particular *FER* (which regulates cell-cell adhesion and mediates signaling from the cell surface to the cytoskeleton via growth factor receptors), *DRD1* (which is a G protein coupled receptor that stimulates adenylate cyclase and helps mediate vascular tone), and *IGF-1* (a potent activator of the AKT signaling pathway and a potent inhibitor of apoptosis) [15–17].

Complex human diseases may be explored using genetic approaches to gain insight into the complex functional pathways that characterize disease [18]. Given the polygenic nature of many diseases, in particular, sepsis where ~10% of genes code for immune mediators [11], creating a polygenic risk score to summarize the estimated effects of an individual's genetic variants on the sepsis phenotype might be clinically useful and furthermore identifying genetic variants that are associated with sepsis may provide important biological insights. GWAS, by analyzing a multitude of genetic variants in large patient populations, has the potential to assess many candidate genes for sepsis, determine their relative contributions, and be used to create polygenic risk scores. The primary purpose of this study is to create a polygenic risk score for sepsis and to assess its discrimination. The secondary purpose it to identify those genetic variants most associated with the polygenic risk score.

## Methods

### Study design and setting

This study was approved by the Institutional Review Board (HUM00168165). We used the STREGA checklist when writing our report. All participants had given written informed consent. The Michigan Genomics Initiative (MGI) is a biobank that collects blood samples on adult perioperative patients for later genetic analysis. The MGI cohort and data have been previously described [19]. Briefly, subjects were recruited primarily during surgical encounters at Michigan Medicine and provided consent for linking of their electronic health records and genetic data for research purposes. Samples were genotyped on customized Illumina Human-CoreExome v12.1 bead arrays (Illumina, Inc. San Diego, CA) and subsequently imputed to the Haplotype Reference Consortium using the Michigan Imputation Server [20], providing ~14

million DNA single nucleotide polymorphisms (SNP) with a minimac imputation quality R2>0.3 and minor allele frequency $>10^{-6}$. European ancestry was inferred using principal component analysis with samples from the Human Genome Diversity Panel as known references.

## Outcome measure and data collection

Adult (> 18 years) subjects who were participating in the MGI had their electronic medical record searched for suspected sepsis. Using previously created and validated software [21], sepsis was separately defined using both the Sepsis-2 and Sepsis-3 definitions [5, 6]. Briefly, patients had to have had body fluid (blood, sputum, urine, wound, other) cultures obtained between Jan 28, 2008 and June 26, 2016, received antibiotics started within the 24 hours before or the 72 hours after obtaining cultures, and had ≥ 2 Systemic Inflammatory Response Syndrome (SIRS) or new Sequential Organ Failure Assessment (SOFA) points. Controls were MGI participants who did not meet sepsis criteria and their medical records showed no body fluid cultures with organism growth. Additionally, controls could not have International Classification of Diseases codes for sepsis, SIRS or infection (S1 Table, Fig 1) [22].

## Data analysis

GWAS was performed using Firth bias-corrected logistic regression implemented in the *epacts* software (https://genome.sph.umich.edu/wiki/EPACTS). Univariate logistic regressions were run with each sepsis definition as the dependent variable and each SNP allele dosage as the independent variable. SNPs were assessed for correlations, and for groups of SNPs with correlations greater than 0.5, only one SNP was retained per group. Additionally, one SNP was removed due to lack of variation in predicted genotype, with greater than 99.9% of patients having the same predicted genotype. SNPs with p ≤ 0.01 in the univariate regressions for a sepsis outcome definition were included as independent variables in the respective multivariable logistic regression along with inferred sex, and principal components 1 through 4. Non-parsimonious models were first created to produce the polygenic risk score and the discriminations of the models were assessed as the areas under the receiver operating characteristic curve (c-statistics). Then, forward selections by Akaike Information Criteria, with inferred sex and principal components 1–4 forced to remain in the models, were performed on each multivariable model to find a parsimonious model and the c-statistics were calculated. Genetic variants with final p<0.05 and 95% confidence intervals that excluded 1 were deemed significant [23–25]. *Post hoc*, we conducted a similar regression analysis to determine the variants associated with 90-day mortality in patients with sepsis. Regression and correlation analyses were run using RStudio version 1.4.1103.

## Results

We had 2261 patients with sepsis: 2040 (90%) had Sepsis-2, 1295 (57%) had Sepsis-3 (1074 (47%) had both Sepsis-2 and Sepsis-3), (Fig 2), and 13,068 control patients without sepsis.

Q-Q plots confirmed that the analyses were well-controlled, without an elevated Type I error (S1 and S2 Figs). The Manhattan plots (Figs 3 and 4) showed that we had no variants at univariate p-values $< 5 \times 10^{-8}$. We did, however, find 772 genetic variants univariately associated with Sepsis-2 and 442 associated with Sepsis-3 at the p < 0.01 level. After removal of collinear genetic variants, we created a polygenic risk score from the remaining 320 variants associated with Sepsis-2. This score had good discrimination (c-statistic = 0.752 ± 0.005) and the 218 variants associated with Sepsis-3 had similar discrimination (c-statistic = 0.752 ± 0.007). Both scores also had good calibration (Fig 5).

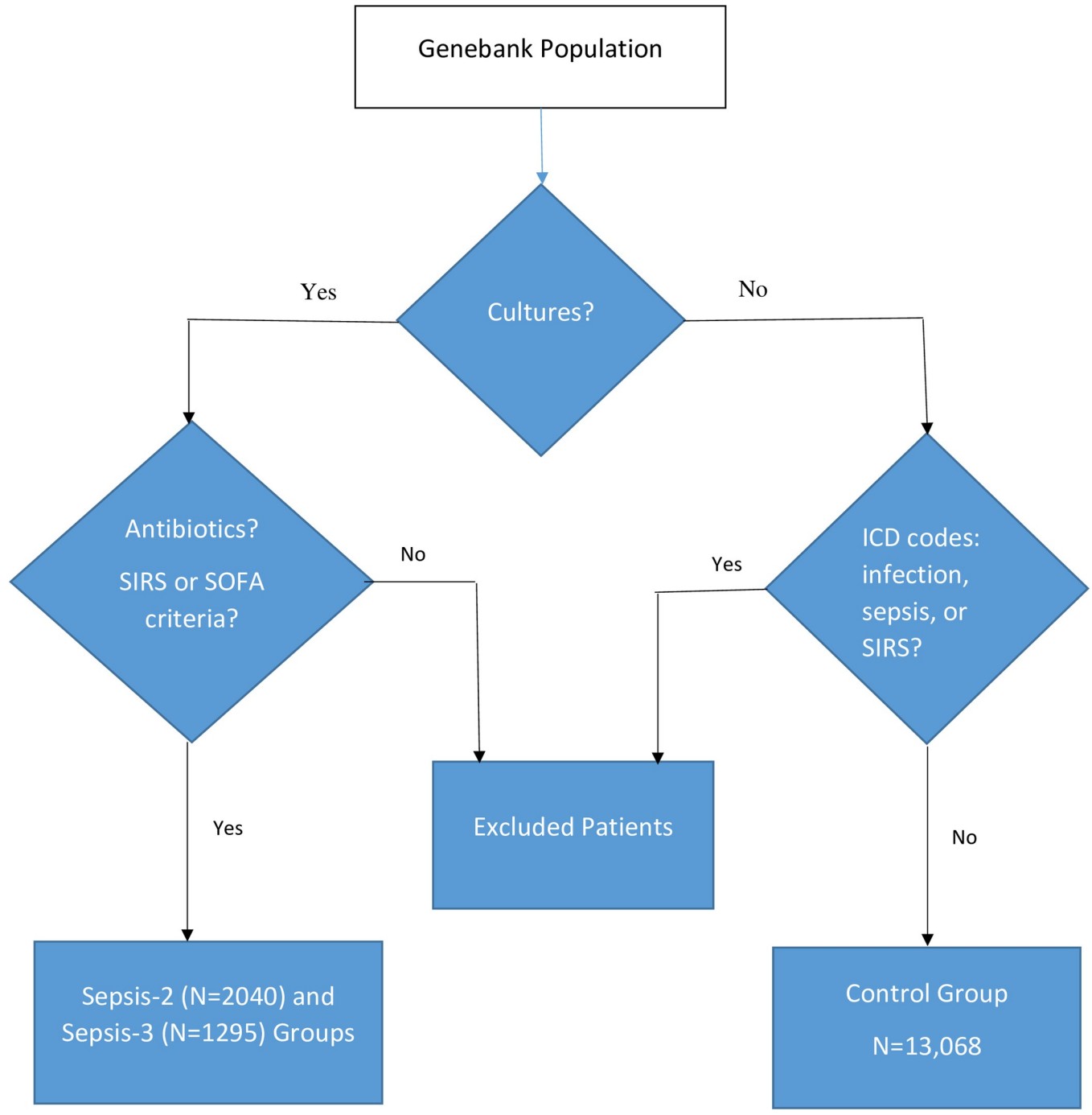

**Fig 1. Patient flowchart.**

After multivariate adjustment to create a parsimonious model, we found 95 variants on 72 genes were independently associated with Sepsis-2 sepsis (S2 Table). Most variates were associated with small increases in the odds ratio of developing sepsis, only five were associated with a doubling or more and five others with at least a halving of the odds ratio compared to the reference gene. A variant in *IL12RB1*, 19:18184213:T:A, had the largest increase in odds ratio (95% confidence interval) = 2.63 (1.57, 4.42), p = 0.0003 and a variant in TJP1, 15:30181207:C:

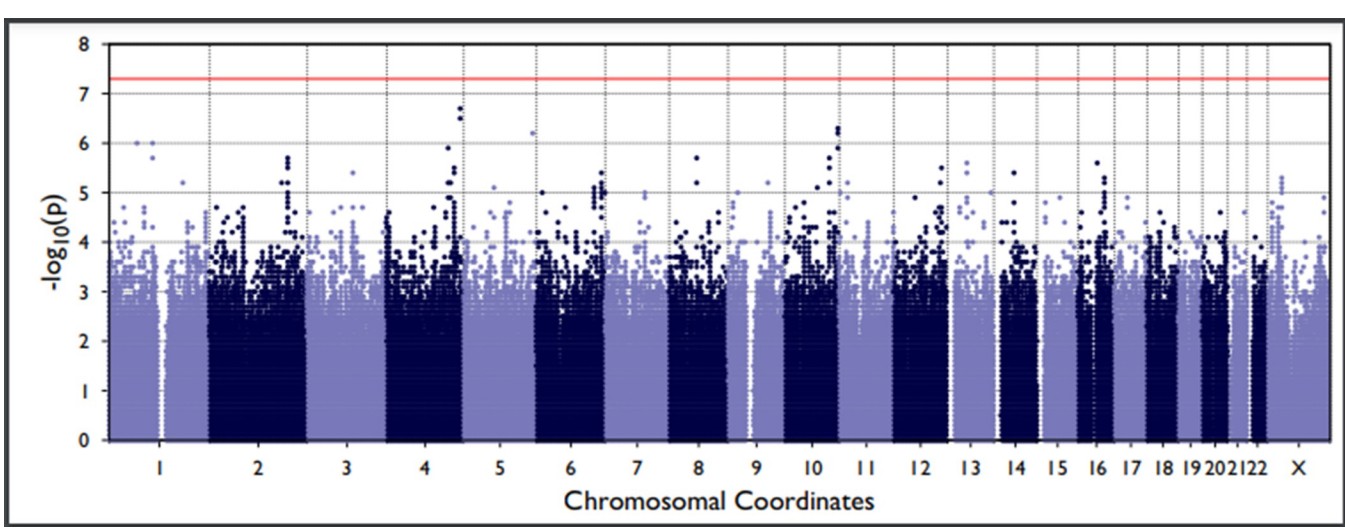

**Fig 2. Venn Diagram showing the number of patients with sepsis by Sepsis-2 (N = 2040) or Sepsis-3 (N = 1295) criteria or both (N = 1074) and the numbers of genes and genetic variants associated with each sepsis phenotype.**

**Fig 3. Manhattan plot showing the single nucleotide polymorphisms and their univariate p-values for Sepsis-2.**

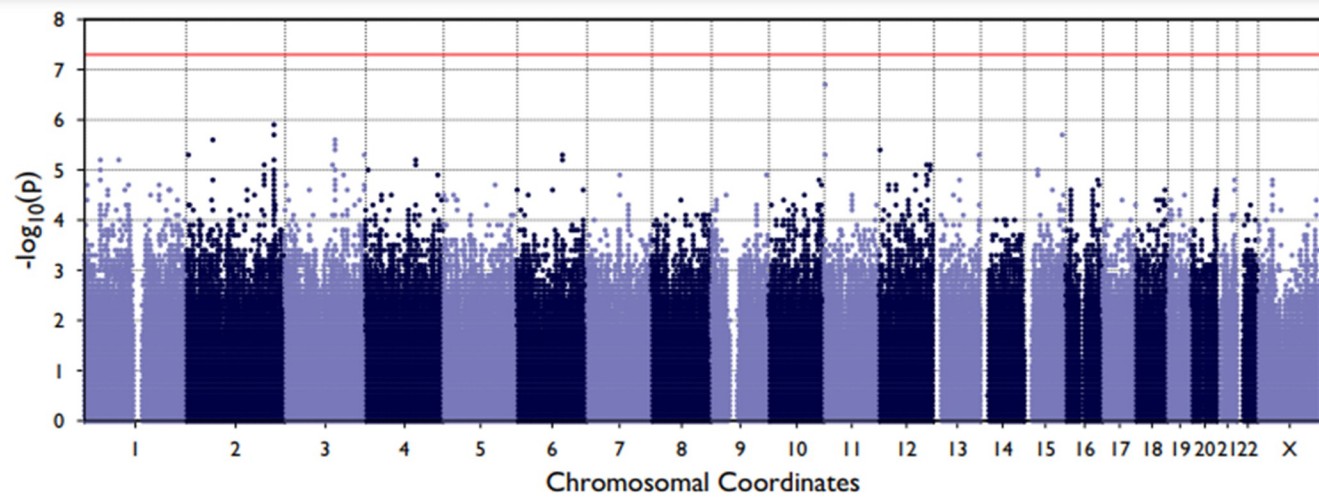

**Fig 4. Manhattan plot showing the single nucleotide polymorphisms and their univariate p-values for Sepsis-3.**

T, had the greatest decrease in odds ratio (95% confidence interval) = 0.29 (0.14, 0.56), p = 0.0003.

In Sepsis-3 patients, after adjustment and reduction, we found 68 variants in 54 genes to be associated with sepsis (S3 Table). Six variants were associated with a more than doubling and four with at least a halving of the odds ratio of sepsis. *PPARA*, 22:46579337:A:T, was associated with the largest increase in odds ratio (95% confidence interval) = 2.30 (1.50, 3.52), p = 0.0001 while *CHRNA7*, 15:32413847:G:A had the greatest decrease in odds ratio (95% confidence interval) = 0.12 (0.02, 0.56), p = 0.008. There were 25 variants that were identically present in both the Sepsis-2 and Sepsis-3 groups out of 32 genes that were present in both groups.

In the mortality analysis, 116 (5%) patients died by 90 days. After adjustment, we found 11 variants on 10 genes to be associated with mortality (S4 Table). The 7:105901555:C:T variant of NAMPT was associated with a 10-fold higher risk of death—odds ratio (95% confidence interval) = 10.61 (2.47, 45.64), p = 0.002.

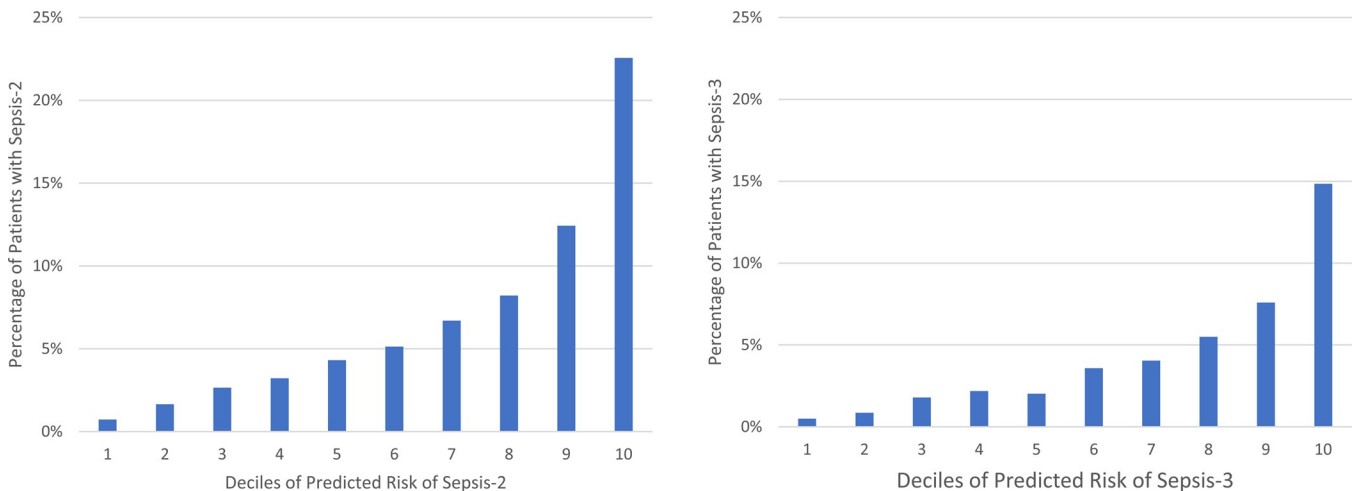

**Fig 5. Calibration plot showing the fraction of patients with sepsis for each decile of risk calculated from the polygenic risk score.** Sepsis-2 (left), Sepsis-3 (right).

## Discussion

We found that the polygenic risk score identifies patients with sepsis with good discrimination. We also found a collection of genetic variants associated with sepsis, either Sepsis-2 or Sepsis-3 and 25 genetic variants associated with both. While Sepsis-2 and Sepsis-3 have distinct definitions, in practice, some patients have both and mortality is higher in patients with both phenotypes than with either one or the other alone [21].

As genotyping becomes increasingly common, the automatic calculation of polygenic risk scores for various diseases by identifying patients at high risk for sepsis, particularly as a complication to other treatment such as chemotherapy, immunotherapy, or surgery, might help physicians better protect these patients or to institute anti-sepsis therapy sooner. For example, a patient at high risk of sepsis undergoing surgery might receive a longer perioperative course of prophylactic antibiotics. Or, a patient receiving immunotherapy for an autoimmune disorder would have a personalized risk of sepsis determined to help understand the risks (infections and sepsis) and benefits of immunotherapy (resolution or amelioration of the disease). This would need to be tested in prospective studies.

We found genes involved in a variety of functions including immune, metabolic, vascular, and signaling. Immune related genetic variants may contribute to the success or lack thereof of clearing the infection, while non-immune genetic variants may contribute to organ damage in sepsis. As many genes have pleomorphic effects, some of these genes are better known for other diseases and have available therapy for them. For example, we found that a PCKS9 variant was associated with a near doubling of the odds of developing either Sepsis-2 or Sepsis-3. PCSK9 inhibitors lower LDL cholesterol and decrease incidence of strokes and myocardial infarction [26]. Two approved inhibitors, alirocumab and evolocumab, are in trials to reduce sepsis-related mortality (NCT03634293 and NCT03869073), respectively. Activation of *PPARA*, a variant we found to be associated with Sepsis-3, inhibits the production of proinflammatory molecules by interfering with macrophages [27]. In a murine model of sepsis, fibrates preserved neutrophil chemotaxis by blocking the effects of lipopolysaccharide [28]. PPARA can be stimulated by fibrates, such as gemfibrozil and fenofibrate, to metabolize lipids. Fibrates lose their ability to inhibit inflammation when *PPARA* gene is knocked out in mice [29]. We found *ANGPT2*, a potent disrupter of microvascular integrity and contributor to vascular leakage, to be associated with both Sepsis-2 and -3. Levels are higher in septic patients and progressively rise in non-survivors. Monoclonal antibodies to *ANGPT2* restore vascular integrity in mice [30].

Previous prospective trials of inhibitors or antibodies to gene products linked to sepsis, such as *IL1RN*, *CD14*, and tissue factor, failed to have benefit, suggesting that these genes are only a small part of a polygenic disease; blocking only this small part would have little overall effect [31–33]. While we didn't find any variants of *IL1RN* or *CD14* to be associated with sepsis, we found two variants of tissue factor pathway inhibitor (*TFPI*), an anticoagulant protein, to be associated with Sepsis-2, but not Sepsis-3.

We also found genes associated with immune function or energy regulation to be associated with mortality. Intracellular NAMPT converts nicotinamide to nicotinamide mononucleotide and is responsible for most of the NAD+ formation in mammals. Administration of NAD + prevents murine septic shock [34]. Extracellular NAMPT activates TLR4, promotes B cell maturation, and inhibits neutrophil apoptosis [35]. Notably, we also found a mutation in CHRNA7 associated with a doubling of the odds of death. Stimulation of this receptor has been associated with inhibited release of tumor necrosis factor and high mobility group box 1, reduced nuclear factor-kappa B activation, and improved survival in a murine model of sepsis [36].

Sepsis presents with various phenotypes [37]. Even for patients who meet Sepsis-2 or Sepsis-3 criteria, patients may have different physiologic or different organs dysfunctions. E.g., one patient may meet Sepsis-2 criteria by having an abnormal heartrate and respiratory rate, while another may have hypothermia and leukopenia. In Sepsis-3, one may have severe isolated renal dysfunction, while another may have hypotension or less severe hypotension plus thrombocytopenia. While some of these differences may be caused by the infectious agent, other differences may be from individual genetic variability. That is, out of the many genetic variants associated with sepsis, each individual will have a subset of these variants, leading to individuals having different sepsis phenotypes. Unfortunately, our study does not have enough patients to investigate this hypothesis.

GWAS in sepsis have potential to improve therapy and outcomes by two methods. First, individualized or precision medicine may be used to tailor therapy based on the particular genetic variants that each individual has [38]. Second, similar to checkpoint inhibitors in cancer chemotherapy that are effective in treating patients with a plethora of diverse mutations, GWAS may help find sepsis checkpoints–particular gene products on which a variety of pathways converge [39, 40].

GWAS analyses typically use very restrictive p values, usually $p < 5 \times 10^{-8}$, to limit the false discovery rate. However, such restrictive p-values only find variants that are relatively common or have relatively large effect sizes [41, 42]. At this threshold, studies of breast cancer using GWAS have missed the association of *BRCA1* and *BRCA2* (prevalences <<1%), even though they are associated with a lifetime risk of breast cancer in women of 65% and 45%, respectively [43, 44]. GWAS to create polygenic risk scores use a much more liberal p threshold and then use a variety of statistical methods to reduce the number of independent variants. A study on multiple sclerosis suggested $p < .2$ and one on schizophrenia and bipolar disorder used a $p < 0.5$ [23, 24]. We chose to use an intermediate threshold, $p < 0.05$, after adjustment. While this produces a large number of variants, mostly with small effect sizes, Boyle et al. have argued that this is explained by genetic networks where the variants are transcribed or activated in relevant tissues [25]. Further studies are needed to determine if, when, where, and how these variants are activated in sepsis.

There are few previous GWAS studies of sepsis. Our study differs from two other studies which assessed the contributions of gene variations and mortality in sepsis [14, 45]. Our findings of different genetic variants may be related to different types of sepsis or patient populations. Our study also differs from a third study, which evaluated prognosis and response to therapy, whereas, our study also assessed genetic variants associated with sepsis occurrence [15]. Our study is more similar to another study, which searched for associations between alleles and sepsis occurrence in extremely premature infants [46]. In their small study with 351 septic infants, they found possible associations with two genes: *FOXC2* and *FOXL1*. We found neither gene to be associated with sepsis in our adult population. There are several differences between their and our study that may explain our failure to reproduce their gene candidates. They studied a multiracial population while we studied a population from European ancestry. While all their cases had positive cultures or meningitis, the physiologic derangements or criteria of sepsis were not described. Neither Sepsis-2 nor Sepsis-3 criteria require positive cultures and most cases of adult sepsis are culture negative [47, 48]. Srinivisan et al. also found separate genes associated with gram positive, gram negative, and fungal organisms [46]. Further study is needed to determine if there are differences in genes associated with culture positive and culture negative sepsis.

In a study of Han Chinese trauma patients, Lu et al. studied 64 genetic variants and found 4 to be associated with sepsis-2: *NOS2*, *PPARG*, *HSPA12A*, and *TLR1* [49]. We found *PPARG* to be associated with only Sepsis-2, similar to their study, giving credence to its association. We

did not find the other 3 genes. This may be related to using different gene candidates in the studies or to differences in ethnicity. Further study is needed.

There are several limitations of this study. First, the analysis was conducted on an inferred mostly European ancestry population. GWAS done on other populations may find other genetic variants related to sepsis. Second, our control population was created on lack of sepsis or infection as documented in the University electronic medical record. As we don't have access to medical records for treatment received at other hospitals, we may have missed episodes of sepsis or infection treated elsewhere. Similarly, control patients may develop sepsis after the analysis. This may bias our results in unknown ways. Further study is needed using patients with more comprehensive records. Third, our GWAS population was selected from patients presenting for surgery with anesthesia at an academic referral medical center. They may not be representative of non-surgical patients or the population at large. Given the constructs of our study, we don't know if patients donated DNA for analysis before or after sepsis. If donated after sepsis, there may be a survival bias in our results. Finally, our study should be validated in other GWAS studies.

The main strength of this study is that the sepsis patients were obtained from careful review of the medical record to find patients who met either Sepsis-2 or Sepsis-3 criteria. Studies of sepsis that use administrative records, such as ICD codes, miss many cases of sepsis, which would introduce biases in GWAS [50, 51].

In conclusion, we found we found that polygenic risk scores predict sepsis with good discrimination. We found 95 variants on 72 genes to be independently associated with Sepsis-2 and 68 variants in 54 genes to be associated with Sepsis-3 sepsis.

## Supporting information

**S1 Fig. Q-Q plots of the single nucleotide polymorphisms for Sepsis-2.** MAF—minor allele frequency.
(PDF)

**S2 Fig. Q-Q plots of the single nucleotide polymorphisms for Sepsis-3.** MAF—minor allele frequency.
(PDF)

**S1 Table. ICD9 codes suggestive of infection that were used to remove possible control patients.**
(DOCX)

**S2 Table. Variants associated with Sepsis-2.** C-statistic = 0.684 (95%confidence interval = 0.672–0.696) ***BOLD*** identify genetic variants associated with both Sepsis-2 and Sepsis-3 sepsis. Factor shows the odds ratios for each of the other components of the analysis. PC–principal component.
(DOCX)

**S3 Table. Variants associated with Sepsis-3.** C-statistic = 0.693 (95%confidence interval = 0.678–0.707) ***BOLD*** identify genetic variants associated with both Sepsis-2 and Sepsis-3 sepsis. Factor shows the odds ratios for each of the other components of the analysis. PC–principal component.
(DOCX)

**S4 Table. Variants associated with mortality.** C-statistic = 0.721 (95% confidence interval = 0.671–0.768). PC–principal component.
(DOCX)

## Acknowledgments

The authors acknowledge the University of Michigan Precision Health Initiative and Medical School Central Biorepository for providing biospecimen storage, management, processing and distribution services and the Center for Statistical Genetics in the Department of Biostatistics at the School of Public Health for genotype data curation, imputation, and management in support of this research.

## Author Contributions

**Conceptualization:** Milo Engoren, Nicholas Douville, Michael D. Maile, Melissa E. Bauer.

**Data curation:** Milo Engoren, Elizabeth S. Jewell, Stephanie Moser.

**Formal analysis:** Milo Engoren, Elizabeth S. Jewell, Michael D. Maile.

**Investigation:** Milo Engoren.

**Methodology:** Milo Engoren, Elizabeth S. Jewell, Nicholas Douville, Stephanie Moser, Melissa E. Bauer.

**Project administration:** Milo Engoren, Elizabeth S. Jewell, Stephanie Moser.

**Resources:** Stephanie Moser.

**Software:** Milo Engoren.

**Supervision:** Milo Engoren.

**Visualization:** Milo Engoren, Elizabeth S. Jewell.

**Writing – original draft:** Milo Engoren, Elizabeth S. Jewell, Nicholas Douville, Michael D. Maile, Melissa E. Bauer.

**Writing – review & editing:** Milo Engoren, Elizabeth S. Jewell, Nicholas Douville, Stephanie Moser, Michael D. Maile, Melissa E. Bauer.

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
