## [Decision Letter · Decision Letter 0]

14 Dec 2021

PONE-D-21-35066Genetic Variants Associated with SepsisPLOS ONE

Dear Dr. Engoren,

Thank you for submitting your manuscript to PLOS ONE. After careful consideration, we feel that it has merit but does not fully meet PLOS ONE’s publication criteria as it currently stands. Therefore, we invite you to submit a revised version of the manuscript that addresses the points raised during the review process.

We look forward to receiving your revised manuscript.

Kind regards,

Yonglan Zheng, Ph.D.

Academic Editor

PLOS ONE

Journal Requirements:

2.Thank you for stating the following in the Competing Interests section: 

(Nicholas Douville was supported by a Foundation for Anesthesia Education and Research (FAER) Mentored Research Training Grant

Michael Maile was supported by the American Diabetes Association.

Milo Engoren has received consulting fee from Aerogen and Masimo.

The other authors have nothing to declare.)

Reviewers' comments:

Reviewer's Responses to Questions

**Comments to the Author**

1. Is the manuscript technically sound, and do the data support the conclusions?

Reviewer #1: Yes

Reviewer #2: Yes

2. Has the statistical analysis been performed appropriately and rigorously? 

Reviewer #1: Yes

Reviewer #2: Yes

3. Have the authors made all data underlying the findings in their manuscript fully available?

Reviewer #1: No

Reviewer #2: Yes

4. Is the manuscript presented in an intelligible fashion and written in standard English?

Reviewer #1: Yes

Reviewer #2: Yes

5. Review Comments to the Author

Reviewer #1: Engoren et al study here the genetic variants associated with two international consensus definitions of sepsis (Sepsis 2 and Sepsis 3).To that end they analyze by GAWS multiple genetic variants associated with sepsis from stored DNA of > 18 years old adult patients undergoing surgery at Michigan University Medical Center. Electronic medical records were searched for sepsis using the sepsis-2 and sepsis-3 definitions. In addition microbiological cultures, antibiotics use and �2 SIRS or SOFA points were also recorded. Controls did not fulfill the International Classification of Diseases Code for sepsis, SIRS or infection. The authors found 25 genetic variants present in both the sepsis-2 and sepsis-3 groups. They consider that most genetic variants have small effect sizes but cumulatively the polygenic risk scores have good discrimination.

The manuscript is enticing and the underlying idea of considering sepsis as a polygenic disease like hypertension is very attractive, correct in my opinion and worth exploring further.

Major points

1.My main concern regards the selection of the controls as individuals �18 years that did not develop sepsis. I understand as stated in Material and Methods (Page 6, lines 125) that patients and controls were followed from January 28, 2008 through June 26, 2016. Is that correct? Eight years of follow-up is a period long-enough to detect sepsis predisposition in old individuals (for instance �75 years) but too short for 30 years old individuals that underwent a surgery (perhaps a minor orthopedic surgery) at Michigan University Medical Center. Younger controls might develop sepsis many years after finishing the 8 years follow-up period. Could you clarify this point?

2.Could you compare sepsis and control individuals regarding age, sex and main comorbidities linked to sepsis (diabetes, cancer, HIV, other immunodepressions) in one initial table?. This way the reader could see that both arms, septic and control , are well balanced.

3.Could you analyze the effect of these genetic variants studied on sepsis mortality as others did ?(References 14 and 42)

Reviewer #2: The manuscript is about an interesting and significant topic, the genetic predisposition to sepsis. Presented original research is within scope of the journal. Clear, professional English language is used in the text.

There is appropriate Introduction to show context and own results. Literature is well referenced and relevant. Structure of this manuscript is in line with the PLOS ONE standards; supplementary material containing tables and figures is relevant, high quality, well labeled and described.

Research question is well and precisely defined. The primary purpose of this study was to create a polygenic risk score for sepsis and to assess its discrimination; the secondary purpose it to identify those genetic variants most associated with the polygenic risk score.

Investigation has been performed in agreement to high technical and ethical standards, and methods are described well, with sufficient information to replicate.

In this GWAS study authors analyzed about 14 million SNPs in a group of 2261 patients with sepsis and more than 13.000 controls founding 772 genetic variants associated with Sepsis-2 and 442 with Sepsis-3 criteria. After multivariate adjustment, 100 variants on 85 genes were associated with Sepsis-2 and 69 variants in 54 genes with Sepsis-3; twenty-five variants were present in both sepsis groups. Most variants showed small effect sizes but cumulatively, the polygenic risk scores had good discrimination.

The results of this study could have practical implementation; based on obtained results, polygenic risk scores could be further explored for diagnostic and prognostic purposes. This is important because sepsis is frequent cause of mortality and morbidity with still unexplained determinants.

The authors critically discussed some limitations of the study linked mainly to the selection criteria for studied population (ethnical origin of patients with sepsis, medical history of both patients and controls). The specificity of this study is that the sepsis patients were obtained from careful review of the medical record to find patients who met either Sepsis-2 or Sepsis-3 criteria. That is important because studies of sepsis that use administrative records, such as ICD codes, miss many cases of sepsis, leading to the biases in GWAS.

All underlying data have been provided; they are sufficient, statistically correct and controlled. Contemporary bioinformatics tools have been used adequately and expertly.

Conclusions are well stated, linked to original research question and to supporting results. The main conclusion is that that established polygenic risk scores predict sepsis with good discrimination.

Authors found 95 variants on 72 genes to be independently associated with Sepsis-2 and 68 variants in 54 genes to be associated with Sepsis-3 criteria.

I suggest accepting this manuscript in present form; no modifications are required.

6. PLOS authors have the option to publish the peer review history of their article (what does this mean?). If published, this will include your full peer review and any attached files.

Reviewer #1: No

Reviewer #2: **Yes: **Full Professor Maja Surbatovic, MD, PhD

---

## [Author Response · Author response to Decision Letter 0]

22 Jan 2022

Journal Requirements:

Response: We have made the style changes.

2.Thank you for stating the following in the Competing Interests section: 

(Nicholas Douville was supported by a Foundation for Anesthesia Education and Research (FAER) Mentored Research Training Grant

Michael Maile was supported by the American Diabetes Association.

Milo Engoren has received consulting fee from Aerogen and Masimo.

The other authors have nothing to declare.)

Response: We have added that statement to the competing interest section of the cover letter and have added a statement on requesting the data.

Response: We have done so.

Response: We obtained the data, with permission, from the Michigan Genomics Initiative. We have contact information for the Michigan Genomics Initiative so that others may similarly apply for the data. Data Availability: Data are available from the Michigan Genomics Initiative https://precisionhealth.umich.edu/our-research/michigangenomics/#request.

Reviewers' comments:

Reviewer's Responses to Questions

Comments to the Author

1. Is the manuscript technically sound, and do the data support the conclusions?

Reviewer #1: Yes

Reviewer #2: Yes

2. Has the statistical analysis been performed appropriately and rigorously? 

Reviewer #1: Yes

Reviewer #2: Yes

3. Have the authors made all data underlying the findings in their manuscript fully available?

Reviewer #1: No

Reviewer #2: Yes

Response: We have added this Data Availability: Data are available from the Michigan Genomics Initiative https://precisionhealth.umich.edu/our-research/michigangenomics/#request.

4. Is the manuscript presented in an intelligible fashion and written in standard English?

Reviewer #1: Yes

Reviewer #2: Yes

5. Review Comments to the Author

Reviewer #1: Engoren et al study here the genetic variants associated with two international consensus definitions of sepsis (Sepsis 2 and Sepsis 3).To that end they analyze by GAWS multiple genetic variants associated with sepsis from stored DNA of > 18 years old adult patients undergoing surgery at Michigan University Medical Center. Electronic medical records were searched for sepsis using the sepsis-2 and sepsis-3 definitions. In addition microbiological cultures, antibiotics use and �2 SIRS or SOFA points were also recorded. Controls did not fulfill the International Classification of Diseases Code for sepsis, SIRS or infection. The authors found 25 genetic variants present in both the sepsis-2 and sepsis-3 groups. They consider that most genetic variants have small effect sizes but cumulatively the polygenic risk scores have good discrimination.

The manuscript is enticing and the underlying idea of considering sepsis as a polygenic disease like hypertension is very attractive, correct in my opinion and worth exploring further.

Major points

1.My main concern regards the selection of the controls as individuals �18 years that did not develop sepsis. I understand as stated in Material and Methods (Page 6, lines 125) that patients and controls were followed from January 28, 2008 through June 26, 2016. Is that correct? Eight years of follow-up is a period long-enough to detect sepsis predisposition in old individuals (for instance �75 years) but too short for 30 years old individuals that underwent a surgery (perhaps a minor orthopedic surgery) at Michigan University Medical Center. Younger controls might develop sepsis many years after finishing the 8 years follow-up period. Could you clarify this point?

Response: We agree that controls can later develop sepsis. We have expanded our Limitations to discuss this. It now reads “As we don’t have access to medical records for treatment received at other hospitals, we may have missed episodes of sepsis or infection treated elsewhere. Similarly, control patients may develop sepsis after the analysis. This may bias our results in unknown ways.”

2.Could you compare sepsis and control individuals regarding age, sex and main comorbidities linked to sepsis (diabetes, cancer, HIV, other immunodepressions) in one initial table?. This way the reader could see that both arms, septic and control , are well balanced. 

Response: Unfortunately, the data we received are deidentified and without age, sex, and comorbidities and we are unable to obtain that data. 

3.Could you analyze the effect of these genetic variants studied on sepsis mortality as others did ?(References 14 and 42)

Response: We have added this as Supplemental Table 3. We describe this in the Methods, Results, and Discussion sections.

Reviewer #2: The manuscript is about an interesting and significant topic, the genetic predisposition to sepsis. Presented original research is within scope of the journal. Clear, professional English language is used in the text.

There is appropriate Introduction to show context and own results. Literature is well referenced and relevant. Structure of this manuscript is in line with the PLOS ONE standards; supplementary material containing tables and figures is relevant, high quality, well labeled and described.

Research question is well and precisely defined. The primary purpose of this study was to create a polygenic risk score for sepsis and to assess its discrimination; the secondary purpose it to identify those genetic variants most associated with the polygenic risk score.

Investigation has been performed in agreement to high technical and ethical standards, and methods are described well, with sufficient information to replicate.

In this GWAS study authors analyzed about 14 million SNPs in a group of 2261 patients with sepsis and more than 13.000 controls founding 772 genetic variants associated with Sepsis-2 and 442 with Sepsis-3 criteria. After multivariate adjustment, 100 variants on 85 genes were associated with Sepsis-2 and 69 variants in 54 genes with Sepsis-3; twenty-five variants were present in both sepsis groups. Most variants showed small effect sizes but cumulatively, the polygenic risk scores had good discrimination.

The results of this study could have practical implementation; based on obtained results, polygenic risk scores could be further explored for diagnostic and prognostic purposes. This is important because sepsis is frequent cause of mortality and morbidity with still unexplained determinants.

The authors critically discussed some limitations of the study linked mainly to the selection criteria for studied population (ethnical origin of patients with sepsis, medical history of both patients and controls). The specificity of this study is that the sepsis patients were obtained from careful review of the medical record to find patients who met either Sepsis-2 or Sepsis-3 criteria. That is important because studies of sepsis that use administrative records, such as ICD codes, miss many cases of sepsis, leading to the biases in GWAS.

All underlying data have been provided; they are sufficient, statistically correct and controlled. Contemporary bioinformatics tools have been used adequately and expertly.

Conclusions are well stated, linked to original research question and to supporting results. The main conclusion is that that established polygenic risk scores predict sepsis with good discrimination.

Authors found 95 variants on 72 genes to be independently associated with Sepsis-2 and 68 variants in 54 genes to be associated with Sepsis-3 criteria.

I suggest accepting this manuscript in present form; no modifications are required.

Response: Thank you for your kind comments.

6. PLOS authors have the option to publish the peer review history of their article (what does this mean?). If published, this will include your full peer review and any attached files.

Do you want your identity to be public for this peer review? For information about this choice, including consent withdrawal, please see our Privacy Policy.

Reviewer #1: No

Reviewer #2: Yes: Full Professor Maja Surbatovic, MD, PhD

---

## [Decision Letter · Decision Letter 1]

23 Feb 2022

Genetic Variants Associated with Sepsis

PONE-D-21-35066R1

Dear Dr. Engoren,

We’re pleased to inform you that your manuscript has been judged scientifically suitable for publication and will be formally accepted for publication once it meets all outstanding technical requirements.

Kind regards,

Yonglan Zheng, Ph.D.

Academic Editor

PLOS ONE

Reviewers' comments:

Reviewer's Responses to Questions

**Comments to the Author**

1. If the authors have adequately addressed your comments raised in a previous round of review and you feel that this manuscript is now acceptable for publication, you may indicate that here to bypass the “Comments to the Author” section, enter your conflict of interest statement in the “Confidential to Editor” section, and submit your "Accept" recommendation.

Reviewer #1: All comments have been addressed

2. Is the manuscript technically sound, and do the data support the conclusions?

Reviewer #1: Yes

3. Has the statistical analysis been performed appropriately and rigorously? 

Reviewer #1: Yes

4. Have the authors made all data underlying the findings in their manuscript fully available?

Reviewer #1: Yes

5. Is the manuscript presented in an intelligible fashion and written in standard English?

Reviewer #1: Yes

6. Review Comments to the Author

Reviewer #1: I am pleased with the changes introduced in the revised version of the manuscript and now in my opinion n it might be accepted in PLOSONE

7. PLOS authors have the option to publish the peer review history of their article (what does this mean?). If published, this will include your full peer review and any attached files.

Reviewer #1: No

---

## [Editor Report · Acceptance letter]

3 Mar 2022

PONE-D-21-35066R1 

Genetic Variants Associated with Sepsis 

Dear Dr. Engoren:

I'm pleased to inform you that your manuscript has been deemed suitable for publication in PLOS ONE. Congratulations! Your manuscript is now with our production department. 

Kind regards, 

on behalf of

Dr. Yonglan Zheng 

Academic Editor

PLOS ONE